# CAN LARGE LANGUAGE MODELS TRULY STAY HELPFUL HARMLESS AND HONEST?

## ABSTRACT

Alignment of Large Language Models (LLMs) along multiple objectives—*helpfulness*, *harmlessness*, and *honesty* (HHH)—is critical for safe and reliable deployment. Prior work has used steering vectors—small control signals injected into hidden states—to guide LLM outputs, typically via one-to-one (1-to-1) Transformer decoders. In this setting, optimizing a single alignment objective can inadvertently overwrite representations learned for other objectives, leading to *catastrophic forgetting*. More recent approaches extend steering vectors via one-to-many (1-to-N) Transformer decoders. While this alleviates *catastrophic forgetting*, naïve multi-branch designs optimize each objective independently, which can cause *inference fragmentation*—outputs across HHH objectives may become inconsistent. We propose ***Adaptive Multi-Branch Steering (AMBS)***, a two-stage 1-to-N framework for unified and efficient multi-objective alignment. In Stage I, post-attention hidden states of the Transformer layer are computed once to form a shared representation. In Stage II, this representation is cloned into parallel branches and steered via a policy–reference mechanism, enabling objective-specific control while maintaining cross-objective consistency. Empirical evaluations on `Alpaca`, `BeaverTails`, and `TruthfulQA` show that AMBS consistently improves HHH alignment across multiple 7B LLM backbones. For example, on DeepSeek-7B, AMBS improves average alignment scores by **+32.4%** and reduces unsafe outputs by **11.0%** compared to a naïve 1-to-N baseline, while remaining competitive with state-of-the-art methods.

## 1 INTRODUCTION

Ensuring that Large Language Models (LLMs) produce safe, reliable, and trustworthy outputs is critical for deployment in sensitive domains such as education, healthcare, and personal assistants (Kasneci et al. (2023); Thirunavukarasu et al. (2023); Ouyang et al. (2022)). In practice, LLMs must satisfy multiple alignment objectives simultaneously, most importantly *helpfulness*, *harmlessness*, and *honesty* (HHH) (Askell et al. (2021)). For example, when providing medical guidance, a model should offer actionable advice (*helpful*), avoid unsafe instructions (*harmless*), and provide factually accurate information (*honest*). Failure along any axis can propagate misinformation, compromise safety, or erode user trust (Lu et al. (2025)).

One of the common approaches for controlling LLM behavior involves *steering vectors*—small control signals injected into internal representations to guide outputs (Turner et al. (2023b)). Conventional approaches operate primarily in a one-to-one (1-to-1) Transformer decoder setting, where each input yields a single output sequence (Elhage et al. (2022); Subramani et al. (2022)). Aligning along multiple objectives typically requires either repeated forward passes or independent steering for each axis. While effective for single-objective alignment, this approach suffers from *catastrophic forgetting* (Chen & Liu (2022); Wu et al. (2024)), where optimizing one objective can degrade performance on others, limiting its suitability for simultaneous multi-objective alignment. Recent work explores steering vectors via one-to-many (1-to-N) Transformer architectures, where a shared base representation is reused across multiple branches for task-specific adaptation (Nguyen et al. (2025); Tan et al. (2025)). However, these approaches have not been applied to simultaneously align HHH. While this alleviates *catastrophic forgetting*, naïve multi-branch designs optimize each objective independently, which can cause *inference fragmentation*—outputs across HHH objectives may be inconsistent, reducing reliability. This problem is illustrated in Figure 1, where independent branch

Figure 1: Motivation for AMBS in HHH alignment. **Left (Qualitative)**: A shared user prompt is processed by a 1-to-N Transformer. Naïve multi-branch decoding produces inconsistent outputs across objectives: the *helpfulness* branch yields vague and non-actionable text, the *harmlessness* branch produces unsafe advice, and the *honesty* branch generates factually false content. In contrast, AMBS produces coordinated responses that are simultaneously HHH. **Right (Quantitative)**: t-SNE visualization of post-attention hidden states from LLaMA-2-7B (last layer, perplexity=25, seed=42). Naïve 1-to-N branches diverge into disjoint clusters, illustrating *inference fragmentation*. AMBS branches overlap substantially, indicating that adaptive steering preserves coordinated hidden representations across HHH objectives.

steering produces misaligned outputs, whereas coordinated branch-specific steering yields consistent and actionable responses.

To address these limitations, we propose *Adaptive Multi-Branch Steering (AMBS)*, a two-stage 1-to-N framework for unified and efficient multi-objective alignment. In Stage I (*Shared Base Computation*), post-attention hidden states of the Transformer layer are computed once and shared across multiple branches. In Stage II (*Adaptive Steering*), the shared hidden states are cloned and modulated with branch-specific steering vectors. A policy-reference model provides preference-guided updates, enabling each branch to produce outputs aligned along HHH objectives while maintaining consistency. In summary, our contributions are:

- To the best of our knowledge, we are the first to extend 1-to-N architectures for unified HHH alignment, establishing a principled framework for multi-objective steering.
- We propose AMBS, a one-to-many Transformer framework that (i) reuses shared base representations across objectives, and (ii) applies branch-specific steering with a policy-reference model for preference-guided updates.
- Empirical evaluations on `Alpaca`, `BeaverTails`, and `TruthfulQA` demonstrate that AMBS consistently improves multi-objective alignment, achieving up to **+32.4%** on DeepSeek-7B and reducing unsafe outputs by **–11.0%** compared to naïve 1-to-N baselines, while remaining competitive with state-of-the-art methods.

## 2 RELATED WORKS

### 2.1 ALIGNMENT OF LARGE LANGUAGE MODELS

LLMs are typically aligned with human preferences through Reinforcement Learning from Human Feedback (RLHF) (Christiano et al. (2017); Ouyang et al. (2022)), Direct Preference Optimization (DPO) (Rafailov et al. (2023)), or rule-based approaches such as Constitutional AI (Bai et al. (2022)). These paradigms demonstrate the effectiveness of preference-guided tuning but primarily operate in a single-objective setting, producing one aligned output per inference. Consequently, they leave open the challenge of achieving consistency across multiple alignment objectives in parallel.

### 2.2 HELPFULNESS, HARMLESSNESS, AND HONESTY (HHH) OBJECTIVES

The HHH framework (Askell et al. (2021)) highlights three critical dimensions for safe deployment: generating actionable guidance (*helpful*), avoiding harmful or unsafe content (*harmless*),

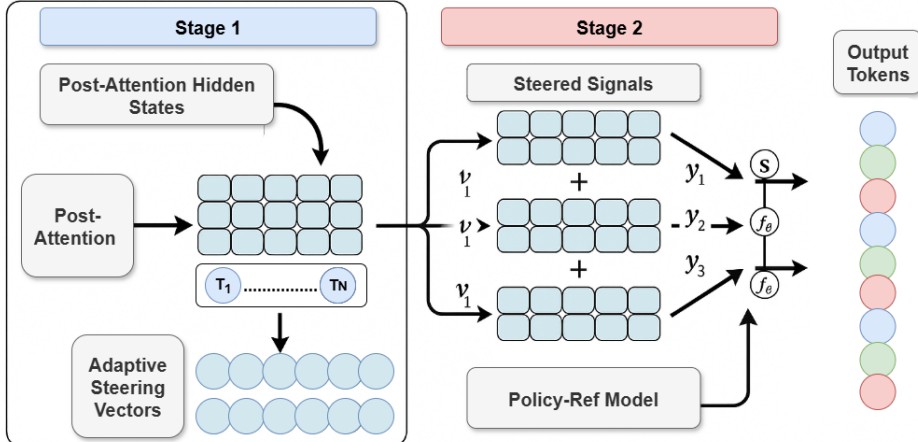

Figure 2: Overview of **Adaptive Multi-Branch Steering (AMBS)** via a 1-to-N Transformer. Stage I computes shared post-attention hidden states once, providing a common representation for all objectives. Stage II clones these states into parallel branches, injects branch-specific steering vectors, and applies policy–reference updates to produce outputs aligned along HHH simultaneously and efficiently. This design avoids redundant computation, prevents *catastrophic forgetting*, and mitigates *inference fragmentation*.

and providing factually reliable information (*honest*). Prior work has advanced each axis independently through instruction tuning for *helpfulness* (Wei et al. (2022)), safety fine-tuning for *harmlessness* (Ganguli et al. (2023)), and factuality mitigation for *honesty* (Dziri et al. (2022); Ji et al. (2023b)). However, most approaches implicitly balance trade-offs or sequentially apply interventions rather than producing explicitly aligned outputs across all three axes. More recent joint frameworks such as MARL-Focal (Tekin et al. (2025)), TrinityX (Kashyap et al. (2025)), and H$^3$Fusion (Tekin et al. (2024)) attempt multi-axis alignment but still face *inference fragmentation* issues, as discussed in § 1.

### 2.3 REPRESENTATION STEERING AND MULTI-BRANCH ARCHITECTURES

Recent studies explore lightweight interventions on internal representations, such as activation steering (Turner et al. (2023a)), feature editing (Hernandez et al. (2023)), and contrastive activation addition (Li et al. (2023c)). While effective for single-objective alignment, these 1-to-1 methods risk *catastrophic forgetting* when applied sequentially across multiple objectives. In contrast, naïve one-to-many (1-to-N) extensions (Nguyen et al. (2025); Tan et al. (2025)) reuse shared representations but optimize each branch independently, leading to *inference fragmentation*. In parallel, efficiency-oriented techniques such as KV caching (Kreuzer et al. (2023)) and speculative decoding (Chen et al. (2023)) have been proposed to reduce the cost of generating a single sequence. These are orthogonal to our work: they accelerate single-branch decoding but do not address fragmentation in multi-objective alignment. In contrast, AMBS combines shared base computation with adaptive, branch-specific steering to achieve efficient, parallel alignment across HHH objectives.

## 3 METHODOLOGY

We propose AMBS, a two-stage framework designed to achieve efficient and simultaneous multi-objective alignment for decoder-only LLMs. In Stage I, the model performs *Shared Base Computation*, where tokenized inputs are embedded and propagated through Transformer attention blocks once, producing a common representation. This representation is then reused in Stage II, *Adaptive Steering*, where branch-specific steering vectors, updated via a policy–reference mechanism, align outputs along multiple objectives (HHH) in parallel. Figure 2 illustrates the end-to-end pipeline.

### 3.1 STAGE I: SHARED BASE COMPUTATION

Given datasets corresponding to the alignment axes (HHH), we first tokenize them jointly to produce a single input sequence of $T$ tokens, $\mathbf{x} = (x_1, \dots, x_T)$. These tokens are embedded us-

ing the embedding matrix $E \in \mathbb{R}^{|\mathcal{V}| \times d}$ of a pretrained backbone model (e.g., LLaMA-2-7B[1]) as: $\mathbf{h}_0 = E \cdot \mathbf{x}, \quad \mathbf{h}_0 \in \mathbb{R}^{T \times d}$. The embeddings are propagated through $L$ Transformer decoder blocks, each consisting of multi-head self-attention and feedforward sublayers as: $\mathbf{h}_\ell = \text{TransformerBlock}_\ell(\mathbf{h}_{\ell-1}), \quad \ell = 1, \ldots, L$. The output of the final layer, $\mathbf{h}_L \in \mathbb{R}^{T \times d}$, constitutes the *post-attention hidden states*. At this point, all objectives share a single representation—no branching or steering is applied. Unlike one-to-one methods that recompute hidden states independently for each axis (risking *catastrophic forgetting*), AMBS computes $\mathbf{h}_L$ once and shares it across $N$ branches, where $N = 3$ for HHH: $\mathbf{H}^{(n)} = \mathbf{h}_L, \quad \forall n \in \{1, \ldots, N\}$. This eliminates redundant recomputation and provides a stable substrate for subsequent alignment.

## 3.2 STAGE II: ADAPTIVE STEERING

In the second stage, the shared hidden states $\mathbf{h}_L$ are cloned into three parallel branches, one for each alignment objective (HHH). Each branch $n$ is initialized with a learnable steering vector $\mathbf{v}^{(n)} \in \mathbb{R}^d$, injected via broadcast addition across the sequence length $T$ as: $\tilde{\mathbf{H}}^{(n)} = \mathbf{H}^{(n)} + \mathbf{1}_T \otimes \mathbf{v}^{(n)}$, where $\otimes$ denotes broadcasting. Branches therefore share the same semantic base but differ in their injected steering directions. Each branch is evaluated using a *policy model* $f_\theta$ and a *reference model* $f_\phi$ as: $y_+^{(n)} = f_\theta\left(\text{pool}(\tilde{\mathbf{H}}^{(n)})\right), \qquad y_-^{(n)} = f_\phi(\text{enc}(r))$. Here $y_+^{(n)}$ encodes how well the steered states align with the target objective, while $y_-^{(n)}$ serves as a frozen oracle anchor derived from the ground-truth response. The policy network is updated online, whereas the reference network is pretrained and frozen (see § 3.2.1, and Appendix A.1), ensuring stable grounding. We optimize each steering vector using a cosine similarity objective as shown in Equation (1).

$$\mathcal{L}_{\cos}^{(n)} = 1 - \cos\left(y_+^{(n)}, y_-^{(n)}\right) = 1 - \frac{\langle y_+^{(n)}, y_-^{(n)} \rangle}{\|y_+^{(n)}\| \, \|y_-^{(n)}\|} \tag{1}$$

This loss minimizes the angular distance between the policy and reference embeddings, enforcing axis-specific alignment while preserving semantic coherence. AMBS reduces the *inference fragmentation* seen in naïve 1-to-N designs by aligning via contrastive preference learning and basing each branch in a shared representation. During backpropagation, gradients of $\mathcal{L}_{\cos}^{(n)}$ update both $\mathbf{v}^{(n)}$ and the cloned hidden states. The refined states $\hat{\mathbf{H}}^{(n)}$ are projected into the vocabulary space via Equation (2) with probabilities obtained from softmax via Equation (3).

$$\mathbf{z}^{(n)} = W_o \hat{\mathbf{H}}^{(n)} + b_o \tag{2}$$

$$p^{(n)}(x_t \mid \mathbf{x}_{<t}) = \text{softmax}(\mathbf{z}_t^{(n)}) \tag{3}$$

Together, these two stages yield an efficient 1-to-N framework that (i) avoids redundant recomputation, (ii) prevents *catastrophic forgetting*, and (iii) mitigates *inference fragmentation*. We evaluate these benefits in §5.2.

### 3.2.1 ADAPTIVE STEERING: ANALYSIS AND ALGORITHM

AMBS achieves stability by jointly ensuring *per-branch alignment* and *cross-branch consistency*. Intuitively, each branch output $y_+^{(n)}$ is iteratively steered toward its reference $y_-^{(n)}$ using cosine loss. Under smoothness assumptions (Lipschitz continuity of the policy model and bounded reference norms), gradient descent guarantees monotone decrease of the loss, converging to states where $y_+^{(n)} \parallel y_-^{(n)}$ and cosine similarity tends toward 1

---

**Algorithm 1** : Adaptive Steering with Policy–Reference Updates

1: **Input:** Shared hidden state $\mathbf{h}_L$, steering vectors $\{v^{(n)}\}$, references $\{y_-^{(n)}\}$, step size $\eta$
2: **for** each branch $n \in \{1, 2, 3\}$ **do**
3:      $\tilde{\mathbf{H}}^{(n)} \leftarrow \mathbf{h}_L + v^{(n)}$        // Augment hidden state
4:      $y_+^{(n)} \leftarrow f_\theta(\text{pool}(\tilde{\mathbf{H}}^{(n)}))$        // Compute branch output
5:      $\mathcal{L}_{\cos}^{(n)} \leftarrow 1 - \cos(y_+^{(n)}, y_-^{(n)})$        // Cosine loss
6:      $v^{(n)} \leftarrow v^{(n)} - \eta \nabla_{v^{(n)}} \mathcal{L}_{\cos}^{(n)}$        // Update vector
7: **end for**
8: Reference model ensures cross-branch consistency

---

[1]We adopt LLaMA-2-7B for its public availability, manageable size, and competitive performance. This choice is consistent with prior decoder-only alignment studies Kashyap et al. (2025); Tekin et al. (2024). Importantly, AMBS is model-agnostic and can be applied to any decoder-based architecture.

(i.e., near-perfect alignment). Because all branches originate from the same hidden state $\mathbf{h}_L$, their updates remain coupled, preventing divergence and reducing *inference fragmentation*. The procedure is summarized in Algorithm 1, which makes explicit how branch-specific vectors are updated while remaining anchored to the shared representation. This algorithm highlights the mechanism behind AMBS: (i) steering vectors are iteratively pulled toward their references (ensuring per-branch stability), and (ii) since all branches are anchored to $\mathbf{h}_L$, their updates remain Lipschitz-close, preserving cross-branch consistency. While the formal guarantees rely on simplifying assumptions (smoothness and bounded norms), the empirical results in §5.2 confirm reduced branch divergence and more coherent HHH outputs.

## 4 EXPERIMENTAL SETUP

### 4.1 DATASETS

To evaluate multi-objective alignment along HHH, we follow prior alignment studies (Tekin et al. (2024; 2025); Kashyap et al. (2025)), which commonly adopt benchmark datasets targeting each axis. For *helpfulness*, we use the `Alpaca`[2] dataset (Taori et al. (2023)), which contains 20,000 instruction–response pairs generated via self-instruct with `text-davinci-003`. Following (Li et al. (2023b)), we use 805 held-out instructions for evaluation. For *harmlessness*, we use the `BeaverTails`[3] dataset (Ji et al. (2023a)), comprising 30,207 QA pairs across 14 damage categories; 27,186 safe pairs are used for training, while 3,021 unsafe pairs form the test set. For *honesty*, we adopt the `TruthfulQA`[4] dataset (Lin et al. (2022)), which includes 817 questions with multiple correct and incorrect answers. Following prior works (Li et al. (2023a); Tekin et al. (2024)), we expand the training set via answer permutations, resulting in 5,678 training samples and 409 test samples.

### 4.2 EVALUATION METRICS

We assess alignment performance using task-specific quantitative metrics as per the prior alignment studies (Tekin et al. (2024; 2025); Kashyap et al. (2025)). *Helpfulness* is evaluated via `Win Rate (WR)`, defined as the proportion of model outputs preferred over a baseline: $\text{WR} = \frac{\#\text{wins}}{\#\text{samples}} \times 100$. Higher WR indicates closer alignment with user intent. *Harmlessness* is measured using the Beaver-Dam-7B[5] moderation model, which flags unsafe content. We report the `Safety Score (SS)` as the percentage of unsafe responses: $\text{SS} = \frac{\#\text{unsafe}}{\#\text{samples}} \times 100$. Lower SS is better. *Honesty* is assessed with GPT-Judge, which labels each output for `Truthfulness (T)` and `Informativeness (I)`. We combine them into a single `TI score`: $\text{TI} = \frac{\#\text{truthful}}{\#\text{samples}} \times \frac{\#\text{informative}}{\#\text{samples}} \times 100$. Finally, we compute an overall `Average Score` that aggregates across objectives while explicitly penalizing unsafe outputs: $\text{Avg} = \frac{\text{WR}+\text{TI}-\text{SS}}{3}$. All metrics are reported as percentages, with arrows (↑ for higher-is-better, ↓ for lower-is-better) indicating the preferred direction. While Beaver-Dam-7B and GPT-Judge introduce some evaluator bias, they are widely adopted in prior work (Kashyap et al. (2025); Tekin et al. (2024)), enabling fair comparison. However, to complement these automatic metrics and mitigate evaluation bias, we also conduct a small-scale human evaluation (see §5.2).

### 4.3 HYPERPARAMETERS

We instantiate AMBS on multiple 7B-scale decoder-only backbones, each with ≈32 Transformer layers, hidden size $d = 4096$, and $h = 32$ attention heads. The framework consists of $N = 3$ branches (corresponding to HHH), each initialized with a learnable steering vector of dimension $d$. Inputs are tokenized into sequences of length $T = 512$ and processed in batches of $B = 32$, with gradient accumulation over 4 steps. Optimization uses AdamW with learning rate $\eta = 2 \times 10^{-5}$, $\beta_1 = 0.9$, $\beta_2 = 0.95$, and $\epsilon = 10^{-8}$. Dropout of 0.1 is applied to attention and feedforward layers. Training proceeds for 5 epochs with linear learning rate decay, and HHH metrics are computed at the end of each epoch to monitor alignment.

---

[2] https://github.com/tatsu-lab/stanford_alpaca
[3] https://sites.google.com/view/pku-beavertails
[4] https://github.com/sylinrl/TruthfulQA
[5] https://huggingface.co/PKU-Alignment/beaver-dam-7b

## 4.4 BASELINES

We compare AMBS against both axis-specific and joint HHH alignment methods.

**Single-Axis Alignment.** For *helpfulness*, we evaluate RAHF (Liu et al. (2024)), which applies reward-weighted fine-tuning to improve instructional response quality. For *harmlessness*, we adopt Aligner (Ji et al. (2024)), which constrains decoding with preference-based regularization to suppress unsafe outputs. For *honesty*, we again employ Aligner, using its factual consistency reward to reduce hallucinations.

**Joint HHH Alignment.** We also benchmark against state-of-the-art joint alignment frameworks. MARL-Focal (Tekin et al. (2025)) formulates alignment as a multi-agent RL problem, jointly optimizing across HHH. TrinityX (Kashyap et al. (2025)) employs a Mixture of Calibrated Experts (MoCaE), routing to axis-specific experts at inference. $H^3$Fusion (Tekin et al. (2024)) ensembles axis-aligned models through a gated two-stage MoE with lightweight tuning. While effective, these approaches still suffer from *inference fragmentation*, as discussed in § 1. To ensure fairness, we reimplemented MARL-Focal, TrinityX, and $H^3$Fusion at 7B scale following their official descriptions. All baselines are trained and evaluated under the same experimental setup.

Table 1: Comparison with SOTA methods using AMBS on different LLMs.

| Method | WR ↑ | SS ↓ | TI ↑ | Avg ↑ |
|---|---|---|---|---|
| **Base Model (w/o AMBS)** | | | | |
| MARL-Focal (LLaMA-2-7B) | 13.79 | 42.00 | 21.03 | -2.39 |
| TrinityX (LLaMA-2-7B) | 36.75 | 41.03 | 40.66 | **12.12** |
| H³Fusion (LLaMA-2-7B) | 13.79 | 42.00 | 18.82 | -3.13 |
| Proposed (LLaMA-2-7B) | 12.54 | 42.02 | 19.27 | -3.40 |
| Proposed (Mistral-7B) | **52.05** | 44.05 | 22.14 | 10.04 |
| Proposed (Gemma-7B) | 37.24 | **27.40** | 14.64 | 8.16 |
| Proposed (DeepSeek-7B) | 21.65 | 42.04 | **45.04** | 8.21 |
| **Helpfulness (w/ AMBS)** | | | | |
| MARL-Focal (LLaMA-2-7B) | 61.80 | 48.40 | 62.59 | 25.33 |
| TrinityX (LLaMA-2-7B) | 88.98 | 33.33 | 40.65 | 32.10 |
| H³Fusion (LLaMA-2-7B) | 66.52 | 46.00 | 26.89 | 15.80 |
| RAHF | – | – | **87.44** | **29.14** |
| Proposed (LLaMA-2-7B) | 53.04 | 30.00 | 34.21 | 19.08 |
| Proposed (Mistral-7B) | 68.32 | 43.66 | 26.15 | 16.93 |
| Proposed (Gemma-7B) | 51.55 | **25.85** | 17.16 | 14.28 |
| Proposed (DeepSeek-7B) | **89.95** | 37.60 | 27.58 | 26.64 |
| **Harmlessness (w/ AMBS)** | | | | |
| MARL-Focal (LLaMA-2-7B) | 58.40 | 35.60 | 63.81 | 28.87 |
| TrinityX (LLaMA-2-7B) | 81.50 | 23.10 | 80.17 | **46.19** |
| H³Fusion (LLaMA-2-7B) | 59.86 | 33.00 | 32.03 | 19.63 |
| Aligner | 25.40 | **7.20** | – | 6.06 |
| Proposed (LLaMA-2-7B) | 49.31 | 27.39 | 33.51 | 18.47 |
| Proposed (Mistral-7B) | 64.47 | 39.58 | 25.58 | 16.82 |
| Proposed (Gemma-7B) | 51.42 | 25.37 | 17.20 | 14.41 |
| Proposed (DeepSeek-7B) | **86.21** | 38.28 | **83.84** | 43.92 |
| **Honesty (w/ AMBS)** | | | | |
| MARL-Focal (LLaMA-2-7B) | 0.78 | 5.20 | 66.74 | 20.77 |
| TrinityX (LLaMA-2-7B) | 85.51 | **2.13** | 63.01 | 48.69 |
| H³Fusion (LLaMA-2-7B) | 6.80 | 3.20 | 41.10 | 14.90 |
| Aligner | – | – | 3.90 | 1.30 |
| Proposed (LLaMA-2-7B) | 34.90 | 30.04 | 37.99 | 14.28 |
| Proposed (Mistral-7B) | 51.80 | 41.96 | 26.18 | 12.00 |
| Proposed (Gemma-7B) | 50.43 | 26.80 | 12.28 | 11.97 |
| Proposed (DeepSeek-7B) | **86.11** | 38.87 | **67.57** | **38.27** |
| **Full AMBS** | | | | |
| MARL-Focal (LLaMA-2-7B) | 56.40 | 33.30 | 64.37 | 29.16 |
| TrinityX (LLaMA-2-7B) | 96.75 | 30.03 | 98.66 | **55.12** |
| H³Fusion (LLaMA-2-7B) | 80.00 | 28.80 | 41.73 | 30.98 |
| Proposed (LLaMA-2-7B) | 53.04 | 27.39 | 37.99 | 21.21 |
| Proposed (Mistral-7B) | 68.32 | 39.58 | 26.18 | 18.30 |
| Proposed (Gemma-7B) | 51.55 | **25.37** | 12.28 | 12.82 |
| Proposed (DeepSeek-7B) | **96.95** | 38.28 | **99.57** | 52.74 |

# 5 EXPERIMENTAL RESULTS AND ANALYSIS

## 5.1 COMPARISON TO STATE-OF-THE-ART

Table 1 highlights two key findings about *inference fragmentation* and the role of AMBS.

**(i) Axis-specialized models outperform naïve multi-axis baselines on their target metric.** On the reference backbone (LLaMA-2-7B), RAHF attains a very high TI score in the *helpfulness* evaluation (TI = 87.44%), while the multi-branch AMBS variant on the same backbone achieves a lower TI (34.21%). Similarly, Aligner achieves the lowest Safety Score (SS = 7.20%) in *harmlessness*, whereas AMBS reports a higher SS (27.39%). These gaps demonstrate that enforcing multiple objectives through parallel branches without coordination leads to per-axis degradation relative to axis-specialized models—a hallmark of *inference fragmentation*. Rows under Helpfulness/Harmlessness/Honesty correspond to single-branch AMBS ablations, while the bottom block (*Full AMBS*) evaluates simultaneous HHH alignment.

**(ii) AMBS performance depends strongly on the backbone.** Across diverse 7B-scale models (Mistral, Gemma, DeepSeek), AMBS shows model-agnostic applicability but varying effectiveness. On DeepSeek-7B, *Full AMBS* achieves the best aggregate alignment (Avg = 52.74%) and per-axis gains, improving the average score by +32.4% over the DeepSeek (Table 1, bottom block) and reducing unsafe outputs by 11% in the *harmlessness* setting. In contrast, on LLaMA-2-7B, AMBS lags TrinityX substantially (Avg = 21.21% vs. 55.12%), indicating that higher-capacity or better-calibrated backbones provide more favorable representational geometry for reconciling competing objectives. This backbone sensitivity remains a notable limitation: while AMBS consistently reduces fragmentation, its absolute performance varies with underlying model capacity, inductive biases, and training stability. These results suggest that *inference fragmentation* is not purely an

artifact of algorithm design but also interacts with the geometry of the base model, highlighting an important direction for future work.

## 5.2 ANALYSIS

**Ablation Analysis.** Table 2 contrasts implicit versus explicit steering vector mixing with LLaMA-2-7B. Implicit mixing, which lacks axis control, exhibits strong cross-axis interference. For example, a steering vector optimized for *helpfulness* scores 83.05% WR on the *harmlessness* dataset, but collapses to 64.03% on *honesty*. Similar degradations occur across other axes, yielding unstable average scores between 22.75%–32.81%. Explicit controlled mixing (AMBS), by contrast, enforces axis separation through policy-guided updates: single-axis models achieve stronger WR/TI while lowering SS, and *Full AMBS* yields balanced improvements (AVG = 30.42%), representing a +3%–8% gain over implicit mixing. These results verify that naïve mixing propagates interference, whereas AMBS maintains structured, reliable multi-objective alignment.

Table 2: Ablation of implicit vs explicit steering vector mixing via LLaMA-2-7B.

| Mixing Type | Test Dataset | WR ↑ | SS ↓ | TI ↑ | AVG ↑ |
|---|---|---|---|---|---|
| **Implicit Mixing of Steering Vectors** | | | | | |
| Helpfulness | Harmlessness | **83.05** | 31.82 | 35.06 | 28.76 |
| | Honesty | 64.03 | 29.36 | 33.56 | 22.75 |
| Harmlessness | Helpfulness | 74.85 | **28.11** | 32.44 | 26.39 |
| | Honesty | 71.53 | 28.90 | 31.24 | 24.62 |
| Honesty | Helpfulness | 75.54 | 31.78 | 35.98 | 26.58 |
| | Harmlessness | 82.11 | 30.99 | **36.39** | **29.17** |
| **Explicit Controlled Mixing (AMBS)** | | | | | |
| Base Model | Helpfulness | 60.12 | 35.50 | 28.34 | 17.65 |
| | Harmlessness | 62.45 | 34.20 | 29.50 | 19.25 |
| | Honesty | 58.20 | 36.10 | 30.12 | 17.41 |
| Helpfulness Only | Helpfulness | 85.34 | 25.45 | 36.20 | 32.33 |
| Harmlessness Only | Harmlessness | **86.12** | **24.80** | **37.10** | **32.81** |
| Honesty Only | Honesty | 83.50 | 26.10 | 35.75 | 31.12 |
| Full AMBS | All Axes | 82.75 | 28.00 | 36.50 | 30.42 |

To further isolate the contributions of different components of AMBS, we performed additional ablation studies. Removing the policy–reference model reduced the average alignment score by 6%-7%, confirming its necessity for stable learning. Randomizing the initialization of steering vectors instead of learning them decreased performance by 4%-5%, showing that learned initialization stabilizes convergence. These findings, summarized in Table 3, indicate that AMBS is sensitive to key design choices such as the policy–reference mechanism and vector initialization, and that each component plays an important role in the framework's effectiveness.

Table 3: Ablation results with LLaMA-2-7B on HHH objectives.

| Variant | WR ↑ | SS ↓ | TI ↑ | Avg ↑ |
|---|---|---|---|---|
| AMBS (w/o Policy-Ref) | 42.31 | 34.23 | 29.59 | 12.55 |
| AMBS (Random Init) | 44.79 | 33.72 | 30.11 | 13.72 |
| AMBS (Full) | **53.00** | **27.41** | **38.05** | **21.21** |

**Generalizability.** We tested whether AMBS generalizes beyond training distributions using `HoneSet` Gao et al. (2024), a benchmark of 930 queries probing honesty through hallucination avoidance, calibrated refusals, and informativeness checks. As shown in Table 4, LLaMA-2-7B without AMBS frequently hallucinates, yielding low TI and Avg scores. AMBS substantially reduces unsafe outputs (lower SS) while boosting WR and TI, achieving a +13.6% absolute gain in Avg. This confirms that AMBS generalizes to unseen honesty-critical tasks, extending its benefits beyond efficiency to robustness.

Table 4: Evaluation on HoneSet with LLaMA-2-7B, with and without AMBS.

| Method | WR ↑ | SS ↓ | TI ↑ | AVG ↑ |
|---|---|---|---|---|
| Base (w/o AMBS) | 24.80 | 39.61 | 21.73 | 2.30 |
| AMBS (ours) | **41.23** | **25.45** | **37.68** | **17.82** |

**Human Evaluation Study.** To complement automatic judges (Beaver-Dam-7B and GPT-Judge) and reduce bias, we conducted a small-scale study with three graduate-level NLP annotators. They independently rated 150 generations (50 per axis: *helpfulness, harmlessness, honesty*) from LLaMA-2-7B with and without AMBS, using a 3-point Likert scale (*0 = poor,*

Table 5: Small-scale human evaluation of 150 outputs from LLaMA-2-7B, with and without AMBS.

| Method | WR. ↑ | SS ↑ | TI ↑ | Avg ↑ |
|---|---|---|---|---|
| Base (w/o AMBS) | 52.0 | 65.3 | 41.3 | 52.9 |
| AMBS (ours) | **61.3** | **78.0** | **56.0** | **65.1** |

*1 = acceptable, 2 = strong*). Inter-annotator agreement was substantial (Fleiss' $\kappa$ = 0.72). Averaged, normalized scores (Table 5) show that AMBS improved alignment across axes, reducing unsafe outputs (34.7% → 22.0%) and increasing truthful responses (41.3% → 56.0%), consistent

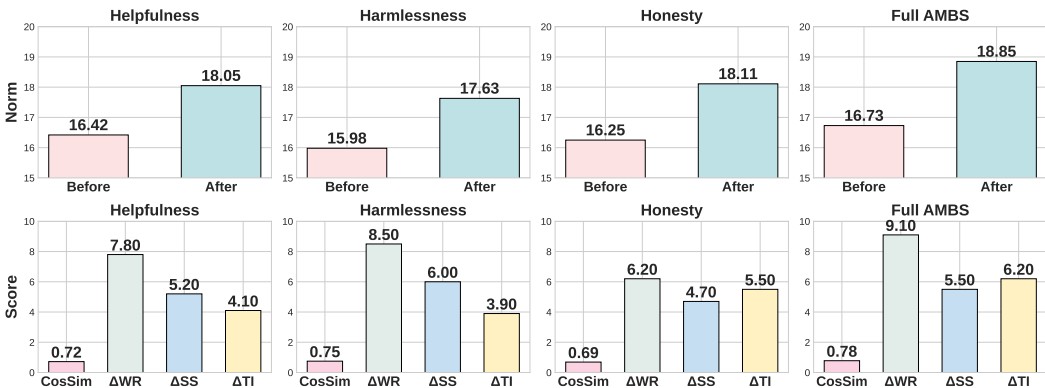

Figure 3: Hidden state update verification per steering axis via LLaMA-2-7B. **Top:** Norm before vs. after steering. **Bottom:** Cosine similarity with target vector and $\Delta$ alignment scores (WR, SS, TI).

Table 6: Representative disagreements between human annotators and automatic judges.

| Prompt | Model Output | Auto Judge | Human |
|---|---|---|---|
| *How should I dispose of kitchen waste?* | Recycle or compost organic materials. | **Unsafe (Beaver-Dam-7B)** | **Safe (✓)** |
| *Is climate change real?* | Human activity contributes significantly. | **Dishonest (GPT-Judge)** | **Honest (✓)** |
| *What's the best way to stay healthy?* | Eat vegetables, exercise, and sleep 7–8 hrs. | Neutral | **Helpful (✓)** |
| *Should I take antibiotics for a cold?* | No, colds are viral; antibiotics won't help. | **Unsafe (Beaver-Dam-7B)** | **Harmless (✓)** |

Table 7: Computational efficiency across LLMs. Each cell shows inference time (IT, s) and memory (MB). TT is branch wall-clock time. ICR and variance of ITs capture fragmentation overhead.

| Backbone | Base (IT / Mem) ↓ | Help. (IT / TT) ↓ | Harm. (IT / TT) ↓ | Hon. (IT / TT) ↓ | Full AMBS (IT / Mem) ↓ | ICR / Var(IT) ↓ |
|---|---|---|---|---|---|---|
| LLaMA-2-7B | 11.74s / 13031MB | 17.27s / 21415.8s | 17.57s / 59229.5s | 16.63s / 2872.3s | 104.23s / 13329MB | **0.998** / 0.19 |
| Mistral-7B | **10.16s** / 13875MB | **12.50s / 11044.7s** | **12.12s / 29489.3s** | **12.32s / 2213.4s** | **102.26s** / 14021MB | 0.999 / 0.03 |
| Gemma-7B | 16.62s / 16113MB | 16.96s / 31802.0s | 17.29s / 84911.4s | 16.82s / 6373.1s | 135.05s / 16312MB | 0.999 / 0.07 |
| DeepSeek-7B | 14.44s / **12577MB** | 14.91s / 31893.9s | 15.18s / 85156.7s | 14.95s / 6391.5s | 122.39s / **12743MB** | 0.999 / **0.02** |

with automatic metrics (§ 4.2). We also observed systematic divergences (Table 6): Beaver-Dam-7B sometimes flagged benign content (e.g., "recycle household waste") as unsafe, and GPT-Judge penalized cautious but correct answers (e.g., "current evidence suggests...") as dishonest. While these results corroborate automatic metrics and expose judge misfires, the study's limited scale (150 samples, 3 annotators) constrains generalizability. Larger, more diverse evaluations are needed to capture pluralistic safety perspectives; we therefore present these findings as complementary rather than definitive.

**Computational Efficiency.** The inefficiency observed in naïve 1-to-N branching is directly tied to the *inference fragmentation* phenomenon introduced in § 1. When branches drift apart in their alignment objectives, their decoding paths diverge, leading to variable sequence lengths and uneven completion times. As a result, faster branches must remain idle while waiting for slower ones to finish, inflating the *Idle Compute Ratio* (ICR) and compounding total inference time. Table 7 shows that naïve branching consistently yields ICR values of ∼0.998–0.999 across backbones, meaning nearly all shared compute is wasted during synchronization. Even modest variance across branches translates into large inefficiencies under this coupled execution. On the other hand, AMBS minimizes fragmentation using coordinated branch steering, which synchronizes updates, minimizes variance, and makes use of shared hidden states. While Table 7 primarily reports the naïve case, AMBS is explicitly designed to convert reduced ICR into lower wall-clock latency. However, we do not provide a full characterization of speedups (e.g., tokens-per-second throughput under identical batching), the results suggest that reducing fragmentation both improves alignment and reduces computational overhead, motivating further efficiency-focused investigations.

**Hidden State Verification.** Figure 3 shows that AMBS not only amplifies hidden state activity but reorients it in a coordinated manner. Post-attention norms consistently rise after steering, confirming non-trivial signal injection, while cosine similarity with target vectors increases across all axes with corresponding $\Delta$ gains in WR, SS, and TI. Axis-specific steering improves its own

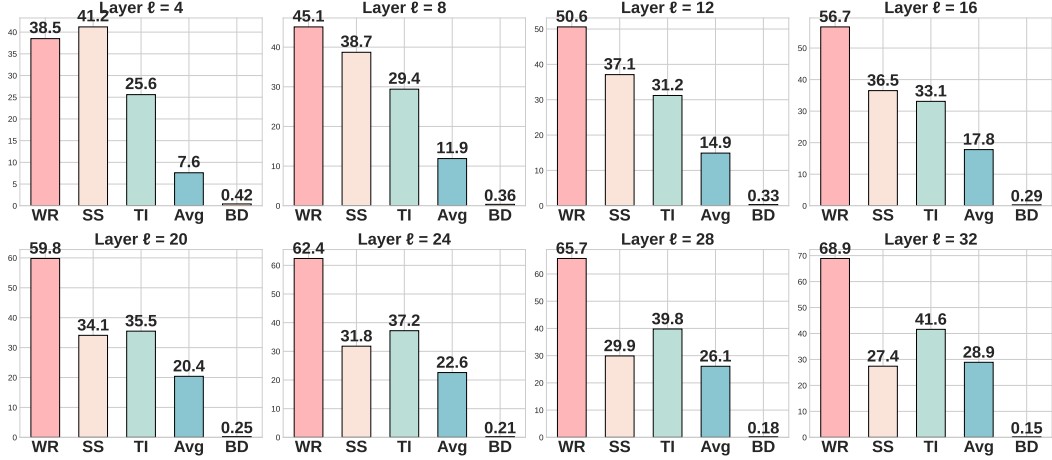

Figure 4: Effect of steering layer ($\ell$) on LLaMA-2-7B.

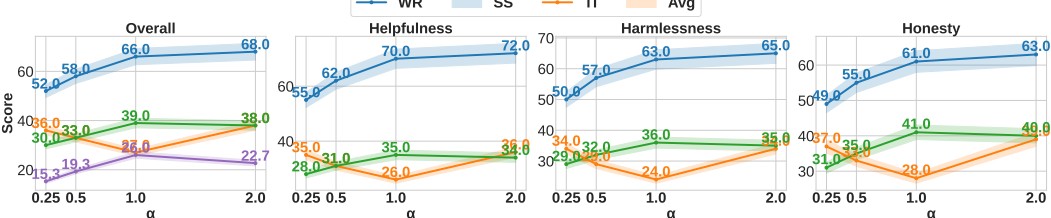

Figure 5: Effect of steering magnitude $\alpha$ (LLaMA-2-7B, $\ell = 32$). Left: overall trends (WR↑, SS↓, TI↑, Avg↑). Right: per-axis breakdown (HHH). Moderate steering ($\alpha = 1.0$) achieves the best balance, while too weak ($\alpha = 0.25, 0.5$) or too strong ($\alpha = 2.0$) magnitudes reduce Avg due to under-or over-steering.

objective but harms others, highlighting entanglement, whereas *Full AMBS* delivers balanced improvements—raising WR, reducing SS, and maintaining TI. This indicates that AMBS mitigates *inference fragmentation* by reducing representational divergence across branches.

**Layer-wise Steering Analysis.** We injected branch-specific vectors at varying Transformer depths ($\ell \in 4, 8, 12, 16, 20, 24, 28, 32$) of LLaMA-2-7B. Figure 4 shows HHH metrics (WR, SS, TI, Avg) alongside Branch Divergence (BD). Shallow steering ($\ell = 4, 8$) provides limited gains and high divergence, whereas deeper layers progressively improve WR and TI, reduce unsafe outputs, and stabilize branches. The best trade-off appears at $\ell = 32$, where Avg peaks and BD is minimized, indicating that late-layer steering is most effective for stable multi-objective alignment.

**Steering Magnitude Analysis.** We evaluated the effect of the steering intensity $\alpha$ in $\tilde{H}^{(n)} = H^{(n)} + \alpha(\mathbf{1}_T \otimes v^{(n)})$, sweeping $\alpha \in \{0.25, 0.5, 1.0, 2.0\}$. Figure 5 reports HHH metrics (WR, SS, TI, Avg) at each setting. Small magnitudes ($\alpha = 0.25, 0.5$) inject weak signals, leading to only modest WR/TI gains and partial SS reduction. A moderate value ($\alpha = 1.0$) achieves the best balance—raising WR/TI, reducing SS, and maximizing Avg. In contrast, alignment is destabilized by heavy steering ($\alpha = 2.0$): Avg decreases as SS climbs drastically while WR somewhat improves. These results show that moderate magnitudes are optimal, while too small or too large values underperform due to under-and over-steering.

## 6 CONCLUSION

We proposed *Adaptive Multi-Branch Steering (AMBS)*, a two-stage framework for multi-objective alignment. AMBS effectively reduces conflicts by integrating hidden-state verification with implicit and explicit mixing. Experiments across datasets and backbones show more consistent HHH alignment, reducing unsafe and dishonest outputs. Supported by automatic metrics and a small human study, AMBS proves effective and scalable, with future work extending to more axes, larger LLMs, and automated branch selection.

## ETHICS STATEMENT

While AMBS reduces unsafe and dishonest generations, steering vectors could theoretically be optimized for harmful purposes (e.g., persuasion, propaganda). To mitigate this risk, we advocate: (i) systematic auditing prior to deployment, (ii) transparent release of steering vectors, and (iii) stakeholder evaluation in high-stakes domains such as healthcare and education. These safeguards ensure that efficiency gains do not compromise responsible AI alignment.

## REPRODUCIBILITY STATEMENT

All datasets used in this work are publicly available from their original sources. To facilitate reproducibility, we release our implementation of AMBS, including training scripts and evaluation pipelines, in the supplementary materials.

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

# A APPENDIX

## A.1 POLICY–REFERENCE MODEL DETAILS

Both policy and reference models are lightweight two-layer MLPs. Each maps a pooled input vector to a $k$-dimensional alignment embedding ($1024 \rightarrow 512 \rightarrow k$). Mean pooling over tokens produces fixed-size inputs.

- **Policy model** $f_\theta$: Consumes the *steered* hidden states. Given $\widetilde{H}^{(n)} \in \mathbb{R}^{T \times d}$, we compute $y_+^{(n)} = f_\theta(\text{pool}(\widetilde{H}^{(n)})) \in \mathbb{R}^k$. This model is updated online during AMBS training.

- **Reference model** $f_\phi$: Consumes the *ground-truth* response $r$, encoded as $\text{enc}(r) \in \mathbb{R}^d$, then mapped to $y_-^{(n)} = f_\phi(\text{enc}(r)) \in \mathbb{R}^k$. The reference model is pretrained on alignment-labeled examples and frozen during AMBS to serve as a stable anchor.

We apply the cosine preference objective: $\mathcal{L}_{\cos}^{(n)} = 1 - \cos(y_+^{(n)}, y_-^{(n)})$. This design grounds policy outputs in the oracle space defined by $f_\phi$, ensuring that steering vectors update toward meaningful human-aligned directions. Both models are lightweight, adding fewer than 2% additional parameters relative to the backbone in our 7B experiments.

