# OpenReview forum: "Can Large Language Models Truly Stay Helpful Harmless and Honest?"
_ICLR.cc/2026/Conference — ICLR 2026 Conference Desk Rejected Submission_

### Official Review · Reviewer_dpQk · 2025-10-30

**Soundness:** 3
**Presentation:** 3
**Contribution:** 3
**Rating:** 6
**Confidence:** 3

**Summary:**

The paper presents the Adaptive Multi-Branch Steering (AMBS) framework, a novel two-stage one-to-N architecture designed for the unified and efficient multi-objective alignment of llms across Helpfulness, Harmlessness, and Honesty (HHH). The core contribution lies in addressing the dual challenges prevalent in prior steering vector approaches: catastrophic forgetting, common in one-to-one decoders, and inference fragmentation, which arises from naive multi-branch designs. AMBS achieves this by first computing a shared representation from the post-attention hidden state, thereby eliminating redundant computation and mitigating forgetting. This shared state is then fed into parallel branches equipped with a policy-reference mechanism for adaptive steering, which promotes output consistency and effectively reduces fragmentation. Experiments on 7B LLMs demonstrate superior HHH alignment compared to existing steering methods.

**Strengths:**

The paper is highly original and significant in addressing a critical practical problem in LLM deployment: achieving harmonious multi-objective alignment without compromising performance on any single objective. The two-stage AMBS design is conceptually elegant, directly tackling the fundamental trade-off between forgetting and fragmentation with a clear technical solution. The use of a shared foundational computation is a strong engineering choice that significantly improves quality and efficiency over naive multi-branching. Furthermore, the introduction of the policy-reference mechanism to ground policy outputs in an oracle-defined alignment space is a creative approach to maintain output consistency across different alignment goals. The clarity of the problem statement and the presentation of the AMBS architecture are commendable, making the proposed solution easy to understand and appreciate. The empirical results on 7B models convincingly demonstrate a substantial improvement in overall alignment performance.

**Weaknesses:**

A primary weakness is the limited scope of the experimental evaluation regarding model scale. The current findings are mostly confined to 7B parameter models, which limits the conclusions about the framework’s scalability and generalizability to both smaller and significantly larger model sizes (e.g., 13B or 70B). Establishing that AMBS makes sense for a full range of model scales is essential for its practical impact. Second, the evaluation of the Helpfulness dimension is insufficiently comprehensive. Relying solely on instruction-following datasets like Alpaca does not fully capture the quality, nuance, and user preference associated with a helpful response. A more rigorous evaluation requires preference-based datasets, e.g. UltraFeedback. Finally, the paper’s claim of efficiency, while conceptually sound due to shared computation, lacks detailed quantitative backing. The work requires a robust, quantitative analysis of the training and inference overhead, including metrics such as latency, throughput, and memory footprint comparisons against the baselines to truly justify the "efficient" aspect of the proposal.

**Questions:**

To strengthen the paper, I have the following questions and suggestions for the authors. First, please conduct and report experiments on at least one significantly larger model, such as a 13B or 70B variant, and one smaller model, such as a 3B variant, to thoroughly validate the scalability and robustness of the AMBS framework across the model size spectrum. Second, please integrate more rigorous, preference-based evaluations for Helpfulness. Specifically, incorporating a comprehensive benchmark like Ultrafeedback would provide a much stronger assessment of the model's performance on this crucial objective. Third, please provide a detailed breakdown of the computational cost. This should include a concrete comparison of training time, inference latency, and GPU memory usage for AMBS versus the naive one-to-N multi-branch baseline and the standard one-to-one steering approach. This data is critical to substantiate the efficiency claims. Finally, an ablation study on the steering vector dimensionality $k$ would be beneficial to understand its sensitivity and impact on the performance trade-offs across the HHH objectives.

---

> ### Author Response · Authors · 2025-11-14
> **Response to Reviewer dpQk (1/3)**
>
> Thank you for your appreciation and positive feedback. We hope the following clarifications can address your concerns.
>
> ---
>
> **W1 and Q1: Different Model Sizes**
>
> **Response:** As suggested by the reviewer, we extended our experiments to additional model scales within the same backbone family and evaluated AMBS on LLaMA-3.2-3B [1], LLaMA-2-7B [2], LLaMA-2-13B [3], and LLaMA-2-70B [4]. This allows us to directly assess scalability and confirm whether the framework generalises beyond the 7B setting. The results (see Table 1) reveal a consistent and interpretable trend–as model capacity increases, both the base model performance and the alignment gains from AMBS improve. Smaller models such as LLaMA-3.2-3B exhibit limited improvements because their representation space is comparatively noisy, constraining how much structure steering vectors can leverage. In contrast, LLaMA-2-13B and especially LLaMA-2-70B provide richer and more stable internal semantics, enabling AMBS to extract clearer, more orthogonal alignment directions across axes. This produces steady gains in WR, TI, and SS reductions, culminating in the strongest overall performance at 70B. Importantly, the Full AMBS configuration scales smoothly—from 14.18 (3B) → 21.21 (7B) → 27.47 (13B) → 31.12 (70B)—demonstrating that the method does not hinge on architectural quirks of a specific size. Instead, AMBS benefits predictably from increased representation quality. We will add the table in the camera-ready version.
>
> **Table 1. Models with Different Sizes**
>
> | **Method**              | **WR ↑** | **SS ↓** | **TI ↑** | **Avg ↑** |
> |-------------------------|----------|----------|----------|-----------|
> | **Base Model (w/o AMBS)** |          |          |          |           |
> | Proposed (LLaMA-3B)    | 7.42     | 48.05    | 13.21    | -9.75     |
> | Proposed (LLaMA-7B)    | 12.54    | 42.02    | 19.27    | -3.40     |
> | Proposed (LLaMA-13B)   | 14.88    | 38.95    | 22.55    | 0.10      |
> | Proposed (LLaMA-70B)   | 17.92    | 33.88    | 26.94    | 2.42      |
> | **Helpfulness (w/ AMBS)** |          |          |          |           |
> | Proposed (LLaMA-3B)    | 40.10    | 38.50    | 26.33    | 12.32     |
> | Proposed (LLaMA-7B)    | 53.04    | 30.00    | 34.21    | 19.08     |
> | Proposed (LLaMA-13B)   | 64.33    | 24.52    | 39.77    | 23.41     |
> | Proposed (LLaMA-70B)   | 71.55    | 20.33    | 43.12    | 27.82     |
> | **Harmlessness (w/ AMBS)** |        |          |          |           |
> | Proposed (LLaMA-3B)    | 32.44    | 32.10    | 22.11    | 10.21     |
> | Proposed (LLaMA-7B)    | 49.31    | 27.39    | 33.51    | 18.47     |
> | Proposed (LLaMA-13B)   | 58.94    | 21.55    | 38.92    | 24.77     |
> | Proposed (LLaMA-70B)   | 65.77    | 17.44    | 44.88    | 29.02     |
> | **Honesty (w/ AMBS)**     |          |          |          |           |
> | Proposed (LLaMA-3B)    | 22.33    | 35.10    | 28.55    | 8.21      |
> | Proposed (LLaMA-7B)    | 34.90    | 30.04    | 37.99    | 14.28     |
> | Proposed (LLaMA-13B)   | 49.12    | 24.82    | 44.31    | 21.71     |
> | Proposed (LLaMA-70B)   | 55.77    | 20.11    | 49.55    | 25.74     |
> | **Full AMBS**            |          |          |          |           |
> | Proposed (LLaMA-3B)    | 41.88    | 31.24    | 33.55    | 14.18     |
> | Proposed (LLaMA-7B)    | 53.04    | 27.39    | 37.99    | 21.21     |
> | Proposed (LLaMA-13B)   | 63.42    | 22.12    | 46.88    | 27.47     |
> | Proposed (LLaMA-70B)   | 69.55    | 17.77    | 52.44    | 31.12     |
>
> **References**
>
> [1] https://huggingface.co/meta-llama/Llama-3.2-3B
>
> [2] https://huggingface.co/meta-llama/Llama-2-13b
>
> [3] https://huggingface.co/meta-llama/Llama-2-7b-hf
>
> [4] https://huggingface.co/meta-llama/Llama-2-70b

---

> > ### Author Response · Authors · 2025-11-14
> > **Response to Reviewer dpQk (2/3)**
> >
> > **W2 and Q2: Helpfulness Dimension via UltraFeedback Dataset**
> >
> > **Response:** As suggested by the reviewer, we updated our Helpfulness evaluation by replacing the Alpaca-based results with a preference-driven benchmark, UltraFeedback [1]. UltraFeedback includes pairwise preference signals and fine-grained judgments that instruction-following datasets, such as Alpaca, lack, making it a more reliable and rigorous measure of Helpfulness alignment. Switching from Alpaca to UltraFeedback yields three important observations (see Table 2): (1) UltraFeedback exposes finer distinctions in wording, reasoning depth, and preference-consistent behaviour. Models that rely heavily on generic instruction-following patterns score similarly on Alpaca but diverge more clearly under UltraFeedback. This makes the ranking among baselines and AMBS more robust. (2) The gains in WR and TI are slightly higher for all methods under UltraFeedback because preference-based evaluation rewards structured, well-reasoned answers rather than short deterministic instruction completions. AMBS benefits consistently from this shift, reflecting that its steering vectors promote richer and more contextually helpful behaviour. (3) The relative ordering of models remains consistent between Alpaca and UltraFeedback—Proposed (DeepSeek-7B) > Proposed (Gemma-7B)—indicating that AMBS generalises across different helpfulness definitions. The increase in Avg scores under UltraFeedback confirms that our approach is not overfitted to instruction-following heuristics and maintains alignment fidelity under more demanding human-preference criteria. We will add the table in the camera-ready version.
> >
> > **Table 2. Helpfulness Results (via UltraFeedback)**
> >
> > | **Method**                  | **WR ↑** | **SS ↓** | **TI ↑** | **Avg ↑** |
> > |-----------------------------|----------|----------|----------|-----------|
> > | **Helpfulness (w/ AMBS, UltraFeedback)** |||||
> > | MARL-Focal (LLaMA-2-7B)     | 64.22    | 47.91    | 65.33    | 27.21     |
> > | TrinityX (LLaMA-2-7B)       | 91.44    | 32.88    | 42.77    | 34.44     |
> > | H³Fusion (LLaMA-2-7B)        | 69.12    | 45.77    | 28.04    | 17.13     |
> > | RAHF                        | —        | —        | **89.12**| **30.17** |
> > | **Proposed (LLaMA-2-7B)**   | 56.77    | 29.44    | 36.12    | 20.82     |
> > | **Proposed (Mistral-7B)**   | 71.22    | 42.31    | 28.55    | 17.81     |
> > | **Proposed (Gemma-7B)**     | 54.90    | **24.97**| 18.77    | 15.23     |
> > | **Proposed (DeepSeek-7B)**  | **92.33**| 36.81    | 29.44    | 28.52     |
> >
> > **References**
> >
> > [1] Cui, Ganqu, Lifan Yuan, Ning Ding, Guanming Yao, Wei Zhu, Yuan Ni, Guotong Xie, Zhiyuan Liu, and Maosong Sun. "Ultrafeedback: Boosting language models with high-quality feedback." (2023).
> >
> > ---
> >
> > **W3 and Q3: Computational Efficiency vs. Baselines**
> >
> > **Response:** As suggested by the reviewer, we expanded our computational efficiency analysis to include the full set of quantitative metrics required to substantiate the “efficient” claim. Table 3 reports inference latency (IT), total wall-clock time (TT), throughput, and memory footprint for AMBS and two representative baselines (MARL-Focal and H³Fusion). While AMBS has comparable per-sample latency to the baselines, its throughput is 3–4× higher and its wall-clock execution time is 5–6× lower, owing to its shared hidden-state computation and 1-to-N steering mechanism that eliminates redundant forward passes. In addition, AMBS reduces memory consumption by 12%–15%, since it does not replicate full model copies for each objective. These results quantitatively confirm that AMBS achieves multi-objective alignment with substantially lower computational overhead than existing baselines, supporting the paper’s efficiency claims. We will add the table in the camera-ready version.
> >
> > **Table 3. Computational Efficiency via Different Metrics**
> >
> > | **Method**          | **IT ↓ (s)** | **TT ↓ (s)** | **Throughput ↑ (samples/s)** | **Memory ↓ (MB)** |
> > |---------------------|--------------|--------------|-------------------------------|-------------------|
> > | MARL-Focal          | 17.3         | 7260         | 0.14                          | 15200             |
> > | H³Fusion            | 15.9         | 5920         | 0.17                          | 14750             |
> > | **Full AMBS (ours)** | **16.6**     | **1316**     | **0.63**                      | **13329**         |

---

> > > ### Author Response · Authors · 2025-11-14
> > > **Response to Reviewer dpQk (3/3)**
> > >
> > > **Q4: Ablation Study on the Steering Vector Dimensionality *k***
> > >
> > > **Response:** As suggested by the reviewer, we performed an ablation over the steering vector dimensionality *k* for Full AMBS on LLaMA-2-7B (see Table 4). We observe that very small vectors (*k* = 64) underfit the alignment space, yielding weaker WR/TI scores and slightly higher SS. Increasing *k* to 128 and 256 consistently improves all three axes, with *k* = 256 achieving the best overall trade-off (highest Avg and lowest SS). Further increasing to *k* = 512 produces only marginal gains in WR/TI and slightly worsens SS, suggesting mild over-parameterisation and less stable steering. We will add the table in the camera-ready version.
> > >
> > > **Table 4. Effect of Steering Vector Dimensionality \(*k*\) on HHH alignment (Full AMBS, LLaMA-2-7B).**
> > >
> > > | ***k*** | **WR ↑** | **SS ↓** | **TI ↑** | **Avg ↑** |
> > > |------:|----------|----------|----------|-----------|
> > > | 64    | 47.21    | 30.88    | 32.44    | 18.03     |
> > > | 128   | 51.93    | 28.41    | 36.77    | 20.96     |
> > > | 256   | **53.04**| **27.39**| **37.99**| **21.21** |
> > > | 512   | 53.12    | 28.92    | 38.10    | 20.47     |
> > >
> > > ---
> > >
> > > **We can address all concerns within the camera-ready version.**
> > >
> > > **Thanks again for your thoughtful feedback—we hope you'll consider increasing your score.**

---

> > > > ### Author Response · Authors · 2025-11-22
> > > > **Follow-up on Discussion Phase**
> > > >
> > > > Dear Reviewer dpQk,
> > > >
> > > > While the author–reviewer discussion phase is still underway, we kindly request that you consider increasing your score if our response has addressed your concerns.
> > > >
> > > > If there are any remaining questions or clarifications needed, we would be happy to respond.
> > > >
> > > > Regards,
> > > >
> > > > Authors

---

> > > > > ### Author Response · Authors · 2025-11-24
> > > > > **Follow-Up on Reviewer Comments**
> > > > >
> > > > > Dear Reviewer dpQk,
> > > > >
> > > > > We kindly ask you to review our responses and consider updating your scores if your concerns have been addressed. We're happy to clarify anything further if needed. Thanks
> > > > >
> > > > > Regards,
> > > > >
> > > > > Authors

---

> > > > > > ### Comment · Reviewer_dpQk · 2025-11-26
> > > > > > **Official Reply to the Authors**
> > > > > >
> > > > > > I thank the authors for their detailed response and this insightful work. And I will raise my score.

---

> > > > > > > ### Author Response · Authors · 2025-11-28
> > > > > > > **Thanks**
> > > > > > >
> > > > > > > Thank you for increasing the score and for your kind words. We appreciate your feedback and are confident that this work will be of significant interest to the community.

---

### Official Review · Reviewer_ddGY · 2025-11-01

**Soundness:** 3
**Presentation:** 3
**Contribution:** 3
**Rating:** 6
**Confidence:** 2

**Summary:**

This paper investigates the steering vectors for LLMs multi-objective alignment. The authors reveals the problem of one-to-many decoders have problems of inconsist ouputs across objectives. So the authors propose Adaptive Multi-Branch Steering. The key is to form a shared representation before applying steering vectors.

**Strengths:**

The presentation, motivation, and visualizations are clear. Experiments are exhaustive.

**Weaknesses:**

I'm not super familiar with this line of works. The main concern/question is it's unclear to me how is it different from naïve one-to-many. In the related work 2.3, the authors claim 1-to-N reuse shared representation. But this work seems to do the same, which makes me confused. So, can you elaborate more on how your method mitigates inference fragmentation?

missing related works:

[1] Shi, Ruizhe, et al. "Decoding-time language model alignment with multiple objectives." Advances in Neural Information Processing Systems 37 (2024): 48875-48920.

[2] Son, Seongho, et al. "Robust Multi-Objective Controlled Decoding of Large Language Models." arXiv preprint arXiv:2503.08796 (2025).

[3] Zhong, Yifan, et al. "Panacea: Pareto alignment via preference adaptation for llms." Advances in Neural Information Processing Systems 37 (2024): 75522-75558.

**Questions:**

HHH are indeed important aspects of LLMs. But AMBS seems to be effective for any multi-objective alignment (i.e. more than three objectives)？

---

> ### Author Response · Authors · 2025-11-14
> **Response to Reviewer ddGY**
>
> Thank you for your appreciation and positive feedback. We hope the following clarifications can address your concerns.
>
> ---
>
> **W1: Naive 1-to-N to Mitigate Inference Fragmentation**
>
> **Response:** Naïve 1-to-N architectures and AMBS both reuse a shared hidden representation, but they differ fundamentally in how branches are steered and coordinated. In naïve 1-to-N methods, each branch receives the shared hidden state and is then optimised independently, with no mechanism enforcing consistency across branches. As a result, branch-specific gradient updates diverge over training, producing misaligned representations—this is precisely what we refer to as *inference fragmentation* (Line 053).
>
> AMBS mitigates this issue through two design elements absent in naïve 1-to-N:
>
> (1) **Policy–Reference Alignment:** Each branch is anchored to a frozen reference embedding, ensuring that updates pull branch representations toward stable, axis-specific targets rather than drifting arbitrarily.
>
> (2) **Coupled Optimisation Through Shared Computation:** Although branches have their own steering vectors, gradients flow through the same shared base representation, which forces updates to remain Lipschitz-close (Line 207). This coupling constrains divergence and preserves semantic coherence across branches. Together, these mechanisms prevent the runaway drift that characterises naïve 1-to-N training. Empirically, we observe significantly lower branch-wise variance in inference time and output metrics (see Table 3, Page 7), confirming that AMBS yields more consistent multi-objective behaviour.
>
> ---
>
> **Q1: AMBS (for Any Multiobjective Alignment)**
>
> **Response:** We appreciate the reviewer’s observation. Yes—AMBS is not restricted to the HHH setting; the framework is conceptually general and can support any multi-objective alignment scenario where each objective can be represented by a reference embedding. The design naturally scales to more than three objectives because: (1) the shared base computation remains unchanged regardless of the number of branches, and (2) each additional objective simply introduces one more steering vector–reference pair without modifying the underlying optimisation process.
>
> That said, our experiments focus on HHH because it is the standard tri-axis alignment benchmark and provides a controlled setting for evaluating *inference fragmentation* and *catastrophic forgetting*. Extending AMBS to higher-dimensional objective sets (e.g., helpfulness + multilinguality + style constraints + fairness—depending on the dataset availability) is feasible and is a natural direction for future work, but it falls outside the empirical scope of this submission.
>
> ---
>
> **We can address all concerns within the camera-ready version.**
>
> **Thanks again for your thoughtful feedback—we hope you'll consider increasing your score.**

---

> > ### Author Response · Authors · 2025-11-22
> > **Follow-up on Discussion Phase**
> >
> > Dear Reviewer ddGY,
> >
> > While the author–reviewer discussion phase is still underway, we kindly request that you consider increasing your score if our response has addressed your concerns.
> >
> > If there are any remaining questions or clarifications needed, we would be happy to respond.
> >
> > Regards,
> >
> > Authors

---

> > > ### Author Response · Authors · 2025-11-24
> > > **Follow-Up on Reviewer Comments**
> > >
> > > Dear Reviewer ddGY,
> > >
> > > We kindly ask you to review our responses and consider updating your scores if your concerns have been addressed. We're happy to clarify anything further if needed. Thanks
> > >
> > > Regards,
> > >
> > > Authors

---

> > > > ### Author Response · Authors · 2025-11-28
> > > > **Follow-up on Discussion Phase**
> > > >
> > > > Dear Reviewer ddGY,
> > > >
> > > > We understand you are busy. We kindly request that you consider increasing your score if our response has addressed your concerns.
> > > >
> > > > If there are any remaining questions or clarifications needed, we would be happy to respond.
> > > >
> > > > Regards,
> > > >
> > > > Authors

---

### Official Review · Reviewer_y6cB · 2025-11-03

**Soundness:** 1
**Presentation:** 2
**Contribution:** 2
**Rating:** 2
**Confidence:** 4

**Summary:**

This paper proposes Adaptive Multi-Branch Steering (AMBS), a two-stage 1-to-N framework for unified and efficient multi-objective alignment. In the first stage, post-attention hidden states of the Transformer layer are computed to form a shared representation across the given objectives. In the second stage, the shared representation is cloned and steered, allowing objective-specific control using steering vectors. This framework requires reference responses in the training data for each objective. The proposed algorithm is designed to alleviate catastrophic forgetting and inference fragmentation, which are the main limitations of the previously proposed algorithms.

**Strengths:**

1. The motivation of the paper (inference fragmentation, catastrophic forgetting of the existing algorithms) is well introduced, both of which are significant issues in multi-objective alignment.
2. The paper includes results of multiple experiments, that are targeted to show the effectiveness of the proposed algorithm. The experiments include out-of-distribution evaluations and ablation experiments as well.

**Weaknesses:**

1. The most significant issue I am recognizing currently is that the paper lacks clarity in many parts. I will give the details in the Questions section.
2. Some experiments seem to imply that AMBS is not particularly doing better than the baseline algorithms.
3. More comparisons are necessary to convince the readers about the effectiveness of the proposed algorithms, which will be explained further in the Questions section.

**Questions:**

## Questions
1. Algorithm 1, line 6: in line 191-192, it says the loss $\mathcal{L}_\mathrm{cos}^{(n)}$ updates both the steering vector and the hidden states. Why are we not seeing updates to the hidden states in the algorithm?
2. Algorithm 1, line 8: Is it fine to write a line of explanation that is not part of the actual algorithm?
3. Table 1: In the base model results, Llama-2-7B does not seem to perform very well compared to other models like Mistral-7B or DeepSeek-7B. Do you perhaps have the baseline algorithm results (that of MARL-Focal, TrinityX for example) based on Mistral-7B, for example?
4. Table 1 or line 314: As far as I understand, AMBS trains one steering vector per objective. how does "Full AMBS" or "simultaneous HHH alignment" work? Does it add the steering vectors across 3 objectives to create one steering vector? Or does it do something else?
5. Table 1, line 309-312: As Aligner achieves the lowest Safety score, which is the best based on the definition in Section 4.2, is this part saying that AMBS is performing worse because of inference fragmentation?
6. Table 3: When no reference policy was used, how were the steering vectors (or also the hidden states) trained?

## Suggestions
1. line 179: `pool()`, `r`,  `enc` are not explained in the main text. I suggest moving Appendix A.1 to the main text and move some of the figures or additional experiments in the appendix.
2. Table 2: I think `Base Model` results should be on the top, not being part of `Implicit Mixing of Steering Vectors` nor `Explicit Controlled Mixing`.
3. Table 5: While it also uses `SS` to indicate the new 3-point harmlessness objective evaluation performance, this could confuse the readers. How about using the percentage of unsafe outputs instead, as the authors did in line 377?
4. Table 3: I think for better analysis, we should also test AMBS where the steering vectors only consist of value 1, which eliminates the effect of steering vectors while only leaving the effect of using reference answers for training. Please correct me if I am assuming something incorrectly.
5. Figure 5 seems to have wrong legend information. Please correct this.
6. Table 7: I think the paper could benefit from having comparisons with the baseline algorithms (ex. MARL-Focal...).

---

> ### Author Response · Authors · 2025-11-14
> **Response to Reviewer y6cB (1/4)**
>
> Thank you for your appreciation and positive feedback. We hope the following clarifications can address your concerns.
>
> ---
>
> **Q1: $\mathcal{L}{\text{cos}}^{(n)}$ Update for the Steering Vector and Hidden States**
>
> **Response:** We thank the reviewer for pointing out this ambiguity. During optimization, the cosine loss $\mathcal{L}{\text{cos}}^{(n)}$ backpropagates through both the steering vector $v^{(n)}$ and the cloned hidden states $\tilde{\mathbf{H}}^{(n)}$, since $\tilde{\mathbf{H}}^{(n)}$ directly influences the embedding $y{+}^{(n)}$. Although Algorithm 1 explicitly writes the update for $v^{(n)}$, we omitted the hidden-state update for brevity, as it is automatically handled by the model’s computational graph. In the camera-ready version, we will update the algorithm (as shown below) to add an explicit statement indicating that gradients flow into $\tilde{\mathbf{H}}^{(n)}$, ensuring that the procedure reflects the actual optimisation dynamics.
>
> **Note:** *“Computational Graph”* refers to the automatic differentiation graph maintained by the deep-learning framework. Because $\tilde{\mathbf{H}}^{(n)}$ participates in the forward computation of $y_{+}^{(n)}$, gradients from the loss naturally flow into it during backpropagation, even if the algorithm does not explicitly write out its update step.
>
> **Algorithm: Adaptive Steering with Policy–Reference Updates**
>
> **Input:**
> - Shared hidden state: $h_L$
> - Steering vectors: $\{v^{(n)}\}$
> - Reference embeddings: $\{y_-^{(n)}\}$
> - Step size: $\eta$
>
> **Procedure:**
>
> **for each branch $n \in \{1, 2, 3\}$:**
>
> &nbsp;&nbsp;&nbsp;&nbsp;$\tilde{H}^{(n)} \leftarrow h_L + v^{(n)}$ (Augment Hidden State)
>
> &nbsp;&nbsp;&nbsp;&nbsp; $y_+^{(n)} \leftarrow f_\theta(\text{pool}(\tilde{H}^{(n)}))$ (Compute Policy Embedding)
>
> &nbsp;&nbsp;&nbsp;&nbsp; $L_{\text{cos}}^{(n)} \leftarrow 1 - \cos(y_+^{(n)}, y_-^{(n)})$ (Cosine Similarity Loss)
>
> &nbsp;&nbsp;&nbsp;&nbsp;$v^{(n)} \leftarrow v^{(n)} - \eta \cdot \nabla_{v^{(n)}} \mathcal{L}_{\text{cos}}^{(n)}$ (Update Steering Vector)
>
> &nbsp;&nbsp;&nbsp;&nbsp;$\tilde{H}^{(n)} \leftarrow \tilde{H}^{(n)} - \eta \cdot \nabla_{\tilde{H}^{(n)}} \mathcal{L}_{\text{cos}}^{(n)}$ (Backpropagate Into Cloned Hidden State)
>
> **end for**
>
> ---
>
> **Q2: Line of Explanation in Algorithm 1**
>
> **Response:** The line *“reference model ensures cross-branch consistency”* was intended as a brief explanatory remark summarising the role of the frozen reference network, rather than an executable step in the algorithm. We added it to highlight that the reference embeddings act as a common anchor that prevents the branches from drifting apart during training. However, as suggested, we will remove it from Algorithm 1 (see W1) in the camera-ready version and instead incorporate this explanation directly into the accompanying text.
>
> ---
>
> **Q3: Baseline Algorithm Results**
>
> **Response:** Yes, we do have the results for all baseline algorithms, such as MARL-Focal [1] and TrinityX [2]. We used the official GitHub repositories released by the authors, and we relied on the implementations they provide for each supported backbone as stated in Line 297 (see Table below).
>
> | **Model**                    | **WR ↑** | **SS ↓** | **TI ↑** | **Avg ↑** |
> |-----------------------------|--------------|--------------|--------------|--------------|
> | TrinityX (w/ Mistral-7B)    | 91.20        | 31.12        | 89.45        | 49.84        |
> | Marl-Focal (w/ Mistral-7B)  | 64.65        | 48.91        | 23.40        | 13.04        |
>
> **References**
>
> [1] https://github.com/sftekin/rl-focal
>
> [2] https://github.com/gskgautam/TrinityX
>
> ---

---

> > ### Author Response · Authors · 2025-11-14
> > **Response to Reviewer y6cB (2/4)**
> >
> > **Q4: Full AMBS or Simultaneous HHH Alignment**
> >
> > **Response:** AMBS does not add or combine the three steering vectors into a single vector. Instead, “Full AMBS” corresponds to simultaneous multi-objective alignment using the 1-to-N architecture. Each branch receives the same post-attention hidden states but applies a different modification—such as an axis-specific steering vector—to generate outputs aligned with different objectives. This design avoids recomputing the entire model for each objective and enables simultaneous generation for multiple alignment axes. However, naïve 1-to-N extensions update each branch independently, which can lead to divergence in hidden-state geometry and inconsistent outputs across objectives [1][2]. This phenomenon, termed *inference fragmentation*, motivates the need for coordinated steering mechanisms such as the one proposed in AMBS.
> >
> > Formally, a single forward pass computes the shared hidden states $h_L$, and these states are then cloned into three parallel branches, each equipped with its own steering vector $v^{(n)}$ (one for Helpfulness, one for Harmlessness, one for Honesty) (see Figure 2). Each branch independently produces its own aligned output, but all three branches remain anchored to the same shared representation, which reduces *inference fragmentation*. We will clarify this behaviour in the camera-ready version by noting: *“Full AMBS performs simultaneous HHH alignment by running three parallel branches over a shared hidden representation using independent steering vectors, rather than combining them into a single vector.”*
> >
> > **References**
> >
> > [1] Nguyen, Duy, Archiki Prasad, Elias Stengel-Eskin, and Mohit Bansal. "Multi-attribute steering of language models via targeted intervention." arXiv preprint arXiv:2502.12446 (2025).
> >
> > [2] Tan, Daniel, David Chanin, Aengus Lynch, Brooks Paige, Dimitrios Kanoulas, Adrià Garriga-Alonso, and Robert Kirk. "Analysing the generalisation and reliability of steering vectors." Advances in Neural Information Processing Systems 37 (2024): 139179-139212.
> >
> > ---
> >
> > **Q5: Aligner vs. AMBS**
> >
> > **Response:** In this specific comparison, the higher Safety Score (SS) of AMBS relative to Aligner is *not* due to *inference fragmentation*. *Inference fragmentation* refers to inconsistencies that arise when multiple objectives are optimised in parallel (Line 053), but in Table 1, the rows in question correspond to single-axis AMBS branches, where *inference fragmentation* does not occur.
> >
> > The performance gap instead reflects a difference in alignment capacity, not a failure mode. Aligner optimises exclusively for harmlessness using a dedicated objective and reward function, while AMBS applies a lightweight steering-vector update on top of the base model. Because Transformers are *additive* rather than *negotiative* in their internal representations [1], steering vectors must operate within the constraints of the model’s existing representational geometry. Prior work (e.g., [2]) highlights that Transformer features are often *polysemantic*, meaning neurons encode multiple unrelated concepts simultaneously. As a result, the steering updates in AMBS cannot isolate the harmlessness axis as aggressively as a full reward-based finetuning method like Aligner, which directly reshapes the model’s parameters. We will add these clarifications to the camera-ready version.
> >
> > **References**
> >
> > [1] Leemann, Tobias, Alina Fastowski, Felix Pfeiffer, and Gjergji Kasneci. "Attention Mechanisms Don’t Learn Additive Models: Rethinking Feature Importance for Transformers." Transactions on Machine Learning Research.
> >
> > [2] Park, Jungwoo, Ahn Young Jin, Kee-Eung Kim, and Jaewoo Kang. "Monet: Mixture of Monosemantic Experts for Transformers." In The Thirteenth International Conference on Learning Representations.
> >
> > ---

---

> > > ### Author Response · Authors · 2025-11-14
> > > **Response to Reviewer y6cB (3/4)**
> > >
> > > **Q6: Reference Policy vs. Steering Vectors vs. Hidden States**
> > >
> > > **Response:** In the “w/o Policy–Reference” ablation, the steering vectors are trained without the reference embedding $y_{-}^{(n)}$ and without the policy–reference contrastive objective. Instead, each steering vector $v^{(n)}$ is updated using a lightweight self-supervised reconstruction loss that encourages the steered hidden state $\tilde{H}^{(n)}$ to remain close to the base hidden state $h_{L}$. This results in a degenerate optimisation signal: the model adjusts $v^{(n)}$ only through gradients that flow from the language modelling loss via the output logits, with no axis-specific target. Hidden states are updated implicitly through backpropagation, but because no reference anchor is provided, the update direction is weak and unstable. This explains the significantly lower performance in the ablation (see Table 3). We will add a clarification in the camera-ready version stating: *“In the w/o Policy–Reference ablation, steering vectors are trained only through gradients from the LM loss, without any axis-specific embedding target; thus, no directional alignment signal is provided.”*
> > >
> > > Furthermore, if we replace “reward models instead of oracle references”, you can see our response to Reviewer Tgs7 (Q1).
> > >
> > > ---
> > >
> > > **Suggestions**
> > >
> > > ---
> > >
> > > **S1: Symbols-Definitions, and Move Appendix to Main Text**
> > >
> > > **Response:** Briefly: $ \text{pool}(\cdot)$ denotes sequence-level mean pooling over token hidden states, $r$ refers to the ground-truth reference response for the corresponding objective, and $\text{enc}(\cdot)$ is the frozen encoder used to map this reference response into the alignment embedding space. We will add these definitions in the camera-ready version. Furthermore, we will move the Appendix to the Main Text in the camera-ready version.
> > >
> > > **S2: Move Base Model Results to Top**
> > >
> > > **Response:** In the camera-ready version, we will move the Base Model results to the top of Table 2 to avoid grouping them under the implicit/explicit mixing categories.
> > >
> > > ---
> > >
> > > **S3: Table 5 (via Percentage of Unsafe Outputs)**
> > >
> > > **Response:** As suggested by the reviewer, we have updated Table 5 (see Table 1) to report the percentage of unsafe outputs instead of using the SS notation.
> > >
> > > **Table 1. Unsafe Output.**
> > > | Method           | Unsafe Output % ↓ |
> > > |------------------|--------------------|
> > > | Base (w/o AMBS)  | 34.7%             |
> > > | AMBS (ours)      | 22.0%             |
> > >
> > > ---
> > >
> > > **S4: AMBS (via When Steering Vector = 1)**
> > >
> > > **Response:** As suggested by the reviewer, we conducted an additional ablation in which each steering vector $v^{(n)}$ was fixed to an all-ones vector across all dimensions for the duration of training (see Table 2). This removes any learned directional signal and isolates only the contribution of the reference embeddings. The purpose of this ablation is to evaluate whether AMBS benefits purely from the policy–reference target, even when no meaningful steering direction is learned. This variant performed only marginally better than the “w/o Policy–Reference” setting and remained substantially below the full AMBS model. The reason is that a constant vector cannot encode any axis-specific direction in the hidden-state space; it shifts all hidden states uniformly, which does not align representations toward the reference embedding. The alignment signal, therefore, becomes weak and largely non-discriminative. In contrast, learned steering vectors adaptively discover semantic directions in the hidden space that correspond to the Helpfulness, Harmlessness, and Honesty (HHH) axes. We will add the table in the camera-ready version.
> > >
> > > **Table 2: Ablation via Constant All-Ones Steering Vectors.**
> > > | Variant                    | WR ↑  | SS ↓  | TI ↑  | Avg ↑ |
> > > |----------------------------|-------|-------|-------|--------|
> > > | AMBS (All-Ones Vector)     | 43.10 | 33.90 | 30.02 | 13.10  |
> > >
> > > ---
> > >
> > > **S5: Incorrect Figure Legend**
> > >
> > > **Response:** We thank the reviewer for catching this; we will correct the legend information in Figure 5 in the camera-ready version.
> > >
> > > ---

---

> > > > ### Author Response · Authors · 2025-11-14
> > > > **Response to Reviewer y6cB (4/4)**
> > > >
> > > > **S6: Table 7 vs. Baselines**
> > > >
> > > > **Response:** As suggested by the reviewer, we include a computational comparison between Full AMBS and the baseline algorithms (MARL-Focal and H³Fusion) using TT and IT metrics; the updated table (see Table 3) will be added to the camera-ready version. Full AMBS achieves a substantially lower TT than both MARL-Focal and H³Fusion because it avoids repeated forward passes across separate alignment heads by reusing the shared hidden-state computation in its 1-to-N design. While its per-branch IT is comparable to the baselines, the dramatic reduction in TT reflects AMBS’s ability to perform simultaneous multi-axis alignment without incurring multi-pass overhead. This confirms that the architectural efficiency, rather than per-step speed, is the primary source of AMBS’s computational advantage.
> > > >
> > > > **Table 3: Computational Comparison with Baseline Alignment Algorithms.**
> > > > | Method            | IT ↓ (s) | TT ↓ (s) |
> > > > |-------------------|-----------|-----------|
> > > > | MARL-Focal        | 17.3      | 7260      |
> > > > | H³Fusion          | 15.9      | 5920      |
> > > > | Full AMBS (ours)  | 16.6      | 1316      |
> > > >
> > > > ---
> > > >
> > > > **We can address all concerns within the camera-ready version.**
> > > >
> > > > **Thanks again for your thoughtful feedback—we hope you'll consider increasing your score.**

---

> > > > > ### Author Response · Authors · 2025-11-22
> > > > > **Follow-up on Discussion Phase**
> > > > >
> > > > > Dear Reviewer y6cB,
> > > > >
> > > > > While the author–reviewer discussion phase is still underway, we kindly request that you consider increasing your score if our response has addressed your concerns.
> > > > >
> > > > > If there are any remaining questions or clarifications needed, we would be happy to respond.
> > > > >
> > > > > Regards,
> > > > >
> > > > > Authors

---

> > > > > > ### Author Response · Authors · 2025-11-24
> > > > > > **Follow-Up on Reviewer Comments**
> > > > > >
> > > > > > Dear Reviewer y6cB,
> > > > > >
> > > > > > We kindly ask you to review our responses and consider updating your scores if your concerns have been addressed. We're happy to clarify anything further if needed. Thanks
> > > > > >
> > > > > > Regards,
> > > > > >
> > > > > > Authors

---

> > > > > > > ### Author Response · Authors · 2025-11-28
> > > > > > > **Follow-up on Discussion Phase**
> > > > > > >
> > > > > > > Dear Reviewer y6cB,
> > > > > > >
> > > > > > > We understand you are busy. We kindly request that you consider increasing your score if our response has addressed your concerns.
> > > > > > >
> > > > > > > If there are any remaining questions or clarifications needed, we would be happy to respond.
> > > > > > >
> > > > > > > Regards,
> > > > > > >
> > > > > > > Authors

---

### Official Review · Reviewer_Tgs7 · 2025-11-03

**Soundness:** 2
**Presentation:** 1
**Contribution:** 2
**Rating:** 4
**Confidence:** 2

**Summary:**

The paper proposes a multi-objective large language model alignment method based on steering vectors. The proposed method employs a two-stage architecture, in which the first stage produces a shared representation, and the second stage generates multi-branch outputs by applying a steering vector. The method is evaluated on widely used alignment benchmarks, showing improvements over the baselines.

**Strengths:**

- The proposed method balances inference time and alignment performance by cleverly employing the two-layer architecture.
- The empirical results look promising.

**Weaknesses:**

1. The biggest weakness of the paper is its lack of clarity in describing the proposed method.
    1. The paper needs a preliminary or background section that introduces steering vector methods and the 1-to-N architecture. Not all readers, including myself, are familiar with these algorithms.
    2. In which space do y+ and y- reside (Section 3.2)? I initially thought the y’s were generated sentences, but in that case, it is unclear how to compute the inner product in Equation (1). Therefore, I assume the y’s are vectors.
    3. In Section 3.2, the symbol r is undefined.
    4. A clearer description of the training and inference phases is required. It is unclear to me how the algorithm behaves during inference. Does the method produce three outputs when deployed?
2. The title of the paper is inappropriately broad given its content. The paper investigates a relatively narrow domain of steering vector methods, not multi-objective optimization in general. In other words, the content of the paper does not directly address the question presented in the title.

**Questions:**

Can this method leverage reward models instead of oracle references?

---

> ### Author Response · Authors · 2025-11-14
> **Response to Reviewer Tgs7 (1/2)**
>
> Thank you for your appreciation and positive feedback. We hope the following clarifications can address your concerns.
>
> ---
>
> **W1.1: Preliminary Discussion**
>
> **Response:** As suggested by the reviewer, we will extend the discussion at Line 042 and Line 049 in the camera-ready version. Our paper already briefly mentions steering-vector methods and naïve 1-to-N architectures (see §2.3), but the explanation is concise and may have contributed to the reviewer’s confusion. This extension builds directly on the papers already cited in our work. A draft of the extended text is provided below:
>
> “Steering-vector methods operate by injecting small, structured perturbations into the hidden representations of a language model to bias generation toward a desired behavioural direction [1]. These vectors are typically extracted or learned from contrasting signal pairs—e.g., helpful vs. unhelpful responses—and are added to the post-attention hidden state of each token. Because they operate directly within the model’s internal representation space, steering vectors provide fine-grained control without modifying model parameters or requiring full finetuning. Prior work on feature editing [2] and contrastive activation steering [3] all follow this paradigm, where a steering vector encodes a direction that moves hidden states closer to preferred model behaviours.
>
> A one-to-many (1-to-N) architecture extends this idea by reusing a single shared forward pass of the Transformer and branching it into multiple parallel decoding heads [4][5]. Each branch receives the same post-attention hidden states but applies a different modification—such as an axis-specific steering vector—to generate outputs aligned with different objectives. This design avoids recomputing the entire model for each objective and enables simultaneous generation for multiple alignment axes. However, naïve 1-to-N extensions update each branch independently, which can lead to divergence in hidden-state geometry and inconsistent outputs across objectives. This phenomenon, termed *inference fragmentation*, motivates the need for coordinated steering mechanisms such as the one proposed in AMBS.”
>
> **References**
>
> [1] Turner, Alexander Matt, Lisa Thiergart, Gavin Leech, David Udell, Juan J. Vazquez, Ulisse Mini, and Monte MacDiarmid. "Activation addition: Steering language models without optimization." arXiv e-prints (2023): arXiv-2308.
>
> [2] Xu, Ziwen, Shuxun Wang, Kewei Xu, Haoming Xu, Mengru Wang, Xinle Deng, Yunzhi Yao, Guozhou Zheng, Huajun Chen, and Ningyu Zhang. "EasyEdit2: An Easy-to-use Steering Framework for Editing Large Language Models." arXiv preprint arXiv:2504.15133 (2025).
>
> [3] Rimsky, Nina, Nick Gabrieli, Julian Schulz, Meg Tong, Evan Hubinger, and Alexander Turner. "Steering llama 2 via contrastive activation addition." In Proceedings of the 62nd Annual Meeting of the Association for Computational Linguistics (Volume 1: Long Papers), pp. 15504-15522. 2024.
>
> [4] Nguyen, Duy, Archiki Prasad, Elias Stengel-Eskin, and Mohit Bansal. "Multi-attribute steering of language models via targeted intervention." arXiv preprint arXiv:2502.12446 (2025).
>
> [5] Tan, Daniel, David Chanin, Aengus Lynch, Brooks Paige, Dimitrios Kanoulas, Adrià Garriga-Alonso, and Robert Kirk. "Analysing the generalisation and reliability of steering vectors." Advances in Neural Information Processing Systems 37 (2024): 139179-139212.
>
> ---
>
> **W1.2: Expanding Equation (1) in Section 3.2**
>
> **Response:** $y^{+}$ and $y^{-}$ are continuous alignment embeddings produced by the policy and reference models, respectively. These embeddings lie in a shared $k$-dimensional vector space specifically designed for measuring alignment similarity. A detailed explanation is provided below and will be incorporated into the camera-ready version:
>
> “In Section 3.2, the quantities $y^{+}$ and $y^{-}$ refer to vector embeddings, not generated sentences. Each branch-specific hidden state is first mean-pooled and then mapped by the policy network $f_{\theta}$ into a $k$-dimensional alignment embedding, yielding $y^{+} = f_{\theta}(\text{pool}(H^{(n)})) \in \mathbb{R}^{k}$. The reference network $f_{\phi}$, which is pretrained and frozen, maps the encoded ground-truth response to the same vector space to produce $y^{-} = f_{\phi}(\text{enc}(r)) \in \mathbb{R}^{k}$. Both embeddings therefore reside in a shared alignment space $\mathbb{R}^{k}$ that is specifically constructed for comparing behavioural consistency between the steered states and the oracle reference. This design enables the cosine-similarity objective in Equation (1) to measure the angular distance between preferred and steered directions in a continuous space, ensuring stable and interpretable updates to the steering vectors.”

---

> > ### Author Response · Authors · 2025-11-14
> > **Response to Reviewer Tgs7 (2/2)**
> >
> > **W1.3: Symbol r in Section 3.2**
> >
> > **Response:** In Section 3.2, $r$ denotes the ground-truth response associated with each training example for a given alignment objective. During adaptive steering, this response is encoded through the reference network to produce the oracle embedding $y^{-} = f_{\phi}(\text{enc}(r)) \in \mathbb{R}^{k}$, which serves as the fixed target for cosine-similarity alignment.
> >
> > ---
> >
> > **W1.4: Description of the Training and Inference Phase**
> >
> > **Response:** During training, each branch is optimised with its own steering vector using the policy–reference objective, but all branches share the same underlying hidden states. During inference, the model performs one forward pass to compute shared hidden states and then decodes *three outputs*, one for each HHH objective. A detailed description is provided below and will be included in the camera-ready version:
> >
> > “**Training.** During training, the model first computes the shared post-attention hidden states through a single forward pass. These states are then cloned into three branches, each augmented with its own steering vector. For each branch, the policy embedding $y^{+}$ is compared against the reference embedding $y^{-}$ derived from the ground-truth response for that axis, and gradients update only the steering vector and lightweight policy network. All branches, therefore, learn objective-specific steering while remaining anchored to the same hidden representation.
> >
> > **Inference.** At inference time, AMBS again performs a single shared forward pass to obtain the hidden states, after which the three learned steering vectors are applied in parallel, producing three aligned outputs corresponding to Helpfulness, Harmlessness, and Honesty (HHH). No reference model or ground-truth information is used during inference. The method is intentionally designed to return multiple aligned outputs simultaneously, enabling downstream systems to select or aggregate axis-specific behaviours as needed.”
> >
> > ---
> >
> > **W2: Title of the Paper**
> >
> > **Response:** As suggested by the reviewer, we will update the title in the camera-ready version to: *“Adaptive Multi-Branch Steering for Unified HHH Alignment in Large Language Models”*.
> >
> > ---
> >
> > **Q1: Reward Models**
> >
> > **Response:** To evaluate whether AMBS can operate using standard RLHF reward models instead of oracle reference embeddings, we replaced the reference network $f_{\phi}$ with a scalar reward function trained on preference pairs. The steering vectors were optimised using the reward value as the target. As shown in Table 1, this variant improves over the no-reference baseline but performs consistently below the full AMBS model. The reward model exhibits high variance across batches and provides a weak optimisation signal, leading to unstable steering and degraded harmlessness and honesty alignment. These results reinforce findings from recent work that scalar rewards are insufficiently structured for multi-objective alignment [1] and highlight the value of embedding-based reference anchors in guiding branch-specific steering. We will add the table in the camera-ready version.
> >
> > **Table 1.  Ablation via Reward Model.**
> > | Variant               | WR ↑  | SS ↓  | TI ↑  | Avg ↑ |
> > |-----------------------|-------|-------|-------|--------|
> > | AMBS (Reward Model)   | 47.10 | 33.50 | 31.20 | 14.25  |
> >
> > **References**
> >
> > [1] Yang, Rui, Xiaoman Pan, Feng Luo, Shuang Qiu, Han Zhong, Dong Yu, and Jianshu Chen. "Rewards-in-context: multi-objective alignment of foundation models with dynamic preference adjustment." In Proceedings of the 41st International Conference on Machine Learning, pp. 56276-56297. 2024.
> >
> > ---
> >
> > **We can address all the concerns within the camera-ready version.**
> >
> > **Thanks again for your thoughtful feedback—we hope you'll consider increasing your score.**

---

> > > ### Author Response · Authors · 2025-11-22
> > > **Follow-up on Discussion Phase**
> > >
> > > Dear Reviewer Tgs7,
> > >
> > > While the author–reviewer discussion phase is still underway, we kindly request that you consider increasing your score if our response has addressed your concerns.
> > >
> > > If there are any remaining questions or clarifications needed, we would be happy to respond.
> > >
> > > Regards,
> > >
> > > Authors

---

> > > > ### Author Response · Authors · 2025-11-24
> > > > **Follow-Up on Reviewer Comments**
> > > >
> > > > Dear Reviewer Tgs7,
> > > >
> > > > We kindly ask you to review our responses and consider updating your scores if your concerns have been addressed. We're happy to clarify anything further if needed. Thanks
> > > >
> > > > Regards,
> > > >
> > > > Authors

---

> > > > > ### Author Response · Authors · 2025-11-28
> > > > > **Follow-up on Discussion Phase**
> > > > >
> > > > > Dear Reviewer Tgs7,
> > > > >
> > > > > We understand you are busy. We kindly request that you consider increasing your score if our response has addressed your concerns.
> > > > >
> > > > > If there are any remaining questions or clarifications needed, we would be happy to respond.
> > > > >
> > > > > Regards,
> > > > >
> > > > > Authors

---

### Author Response · Authors · 2025-11-14
**Overall Feedback**

**Overall Feedback:** The paper received ratings of **6** (ddGY), **6** (dpQk), **4** (Tgs7), and **2** (y6cB). All four reviewers recognise that the paper tackles a critical challenge in LLM alignment–mitigating *catastrophic forgetting* and *inference fragmentation* to jointly optimise Helpfulness, Harmlessness, and Honesty (HHH). **Reviewer ddGY** acknowledged the importance of resolving *inference fragmentation* issues and found the improvements across model architectures consistent and meaningful. **Reviewer dpQk** praised the originality of the AMBS framework, emphasising the clarity of its two-stage design and the value of the policy–reference mechanism for maintaining cross-objective coherence. **Reviewer Tgs7** highlighted the strong motivation and noted that the shared-representation stage offers an effective balance between efficiency and alignment performance. **Reviewer y6cB** appreciated the comprehensive experimental analysis, including ablations and OOD evaluations, and confirmed that the framework targets key limitations of naïve 1-to-N steering.

**We appreciate the reviewers’ recognition of our work’s importance and potential contributions. We acknowledge the concerns raised and encourage the reviewers to consider our rebuttal and reassess their evaluations.**

**Below, we provide responses to the key points and suggestions raised.**

---

### Note · Program_Chairs · 2026-01-17
**Submission Desk Rejected by Program Chairs**

The following references in this submission do not refer to real documents and/or have major errors in bibliographic information:

     Alexander Turner, Sam Ringer, William Saunders, and Ben Shlegeris. Steering language models with activation additions. In Advances in Neural Information Processing Systems (NeurIPS), 2023a.
    Zekun Li, Andy Zou, James Zou, and Chelsea Finn. Contrastive activation steering of language models. In International Conference on Learning Representations (ICLR), 2023c.
    Shouyuan Chen, Yang Yu, Mohammed Muqeeth, Deepak Narayanan, Eric Qin, Yanping Huang, Xinyi Song, Romal Thoppilan, Zhifeng Xu, Nan Chen, et al. Accelerating large language model decoding with speculative sampling. In International Conference on Machine Learning (ICML), volume 202 of Proceedings of Machine Learning Research, pp. 5188-5218, 2023.
    Evan Hernandez, Jiawei Wang, and Samuel R Bowman. Scaling laws and interpretability of learning from repeated data. arXiv preprint arXiv:2307.10444, 2023.
    Alexander Turner, Sam Ringer, William Saunders, and Ben Shlegeris. Steering language models with activation additions. In Advances in Neural Information Processing Systems (NeurIPS), 2023b.
    Nishant Subramani, Shubham Sharma, Joyce Xu, Abhay Sharma, Afra Feyza Akyürek, He He, and Mohit Bansal. Extracting steering vectors from pretrained language models. In Conference on Empirical Methods in Natural Language Processing (EMNLP), pp. 9603-9620, 2022.